# Knockdown of DJ-1 Resulted in a Coordinated Activation of the Innate Immune Antiviral Response in HEK293 Cell Line

**DOI:** 10.3390/ijms25147550

**Published:** 2024-07-10

**Authors:** Keren Zohar, Michal Linial

**Affiliations:** Department of Biological Chemistry, Institute of Life Sciences, The Hebrew University of Jerusalem, Jerusalem 91904, Israel; keren.zohar@mail.huji.ac.il

**Keywords:** cytokine release, interferon, RNA-seq, oxidative stress, dsRNA, siRNA, innate immunity

## Abstract

PARK7, also known as DJ-1, plays a critical role in protecting cells by functioning as a sensitive oxidation sensor and modulator of antioxidants. DJ-1 acts to maintain mitochondrial function and regulate transcription in response to different stressors. In this study, we showed that cell lines vary based on their antioxidation potential under basal conditions. The transcriptome of HEK293 cells was tested following knockdown (KD) of DJ-1 using siRNAs, which reduced the DJ-1 transcripts to only 12% of the original level. We compared the expression levels of 14k protein-coding transcripts and 4.2k non-coding RNAs relative to cells treated with non-specific siRNAs. Among the coding genes, approximately 200 upregulated differentially expressed genes (DEGs) signified a coordinated antiviral innate immune response. Most genes were associated with the regulation of type 1 interferons (IFN) and the induction of inflammatory cytokines. About a quarter of these genes were also induced in cells treated with non-specific siRNAs that were used as a negative control. Beyond the antiviral-like response, 114 genes were specific to the KD of DJ-1 with enrichment in RNA metabolism and mitochondrial functions. A smaller set of downregulated genes (58 genes) was associated with dysregulation in membrane structure, cell viability, and mitophagy. We propose that the KD DJ-1 perturbation diminishes the protective potency against oxidative stress. Thus, it renders the cells labile and responsive to the dsRNA signal by activating a large number of genes, many of which drive apoptosis, cell death, and inflammatory signatures. The KD of DJ-1 highlights its potency in regulating genes associated with antiviral responses, RNA metabolism, and mitochondrial functions, apparently through alteration in STAT activity and downstream signaling. Given that DJ-1 also acts as an oncogene in metastatic cancers, targeting DJ-1 could be a promising therapeutic strategy where manipulation of the DJ-1 level may reduce cancer cell viability and enhance the efficacy of cancer treatments.

## 1. Introduction

Parkinson’s disease (PD) is a neurodegenerative disorder characterized by the progressive loss of dopaminergic neurons, resulting in motor dysfunction and cognitive impairment. Neuronal cell death under oxidative stress and mitochondrial dysfunction are considered major drivers of PD pathologies [1]. DJ-1’s primary role as an essential mitochondrial protein explains much of its role in PD [2]. Mutations in PARK7 (also called DJ-1) were implicated in familial early-onset and sporadic forms of PD [3]. As an oxidative sensor, DJ-1 plays a critical role in maintaining redox homeostasis in all cells by sensing and modulating cellular oxidative responses. DJ-1 is involved in oxidation stress in direct and indirect modes. For example, DJ-1 binds the subunits of mitochondrial complex I. Therefore, upon DJ-1 depletion, mitochondrial function is drastically reduced. Downregulating DJ-1 in neuronal cells led to increased cell death in response to oxidative stress, ER stress, and proteasome inhibition, while overexpression of DJ-1 reduced cell death [4]. Several oxygen-based modifications to the cysteine residues of DJ-1 create a set of modified variants that are partitioned among cellular compartments (mitochondria, cytosol, nucleus, and exosomes) to conduct downstream functions in signaling and transcription regulation [5]. To protect against cell death from various stressors, the modified DJ-1 protein is engaged in a rich protein interaction network that includes often specific signaling pathways and cell types [6,7].

The DJ-1 has garnered considerable interest due to its multifaceted roles in driving numerous processes [1,8]. DJ-1 is involved in transcriptional regulation in direct and indirect modes. To gain insights into the role of DJ-1 in cells, diverse cell systems were used. This included PD-related cell models from humans and rodents, such as primary dopaminergic neurons [9], brain slices, and numerous cell lines [10]. Translocation of DJ-1 to the nucleus and activation of transcription were demonstrated not only in neuronal-like cells (e.g., SH-SY5Y, PC12) [11,12] and numerous primary cell cultures (e.g., astrocytes, microglia, cortical neurons) [13,14,15] but also in routinely used cell lines (HEK293, MIN6, COS7, CHO) [16,17,18]. In expanding the role of DJ-1 in cell signaling, it has been argued that its location, and the pattern of its post-translational modifications, determines the effect of DJ-1 in different cell systems [7], as studied by overexpression [19] or knockdown (KD) (e.g., [16,20]).

Accumulating evidence validates the role of DJ-1 as an oncogene in cancer biology and, more specifically, in tumorigenesis and cancer progression [21]. The expected impact of DJ-1 on cancer is tightly connected to its role in apoptosis, cell proliferation, and metastasis. For example, the role of DJ-1 in suppressing apoptosis in cancer cells allows for the production of the driving oncoproteins, thereby promoting tumor progression. However, the molecular mechanisms underlying DJ-1-mediated modulation of cancer-related pathways remain elusive. In cancer cells, DJ-1 regulates the JNK pathway and consequently impacts autophagy [22,23]. This was further confirmed in prostate cancer [24]. DJ-1, through the regulation of androgen receptor (AR) signaling, affected Beclin1-involved autophagy responses via the JNK-dependent pathway. DJ-1 overexpression reduces LC3 transformation and autophagosome formation, while DJ-1 knockdown has the opposite effect. JNK phosphorylation and Bcl2 dissociation are affected by DJ-1 levels. Recent studies have highlighted the transforming activity of cancer cells. Overexpression of DJ-1 enhances colorectal cancer cell proliferation through the cyclin-D1/MDM2-p53 signaling pathway [25] and acts to increase cancer cell survival through the PI3K-AKT pathway [26]. The elevated levels of DJ-1 have been detected in breast, lung, and prostate cancers, making it a potential biomarker for diagnosis and prognosis [21,27].

DJ-1’s function in the brain has shown its involvement in controlling inflammation [28]. Higher levels of inflammation signatures (including TNF, iNOS, NO, IL-6, cyclooxygenase-2, and p38 phosphorylation) were observed in astrocytes and microglia lacking the DJ-1 mouse model after lipopolysaccharide (LPS) exposure [29]. Similarly, brain slices from DJ-1 knockout mice exposed to IFN-γ exhibited increased levels of IL-6 and TNF, as well as STAT1 phosphorylation [30]. Studies that mimicked glial activation by injecting LPS into the substantia nigra of DJ-1 knockout mice resulted in elevated levels of immune-responsive genes including ICAM-1, IFN-γ, IL-1β, IL-16, IL-17, CXCL11, and more [31]. Reducing DJ-1 levels also accelerated IKK and IkBα phosphorylation, leading to enhanced p65 nuclear translocation in both normal and LPS-exposed cells [6]. The presence of an elevated NF-κB promoter function in DJ-1-depleted cells further confirms DJ-1’s inhibitory function in neuroinflammation [32].

Upon viral infection, the innate system is activated, leading to an antiviral defense mechanism mediated by interferon (IFNs). The type I IFNs (IFNα, IFNβ) activate IFN receptors (IFNARs) via JAK1, TYK2, STAT1, and STAT2 signaling pathways, whereas type II IFN (IFNγ) activates IFNγ receptors (IFNGRs), leading to STAT1 phosphorylation and gene expression. Upon viral infection (e.g., Respiratory syncytial virus; RSV), reactive oxygen species (ROS) levels increase and, as a result of changes in cell redox, the JAK-STAT pathway is activated, resulting in the expression of antiviral genes [33,34]. Elevated ROS signifies the pathogenesis of human diseases such as pulmonary fibrosis and Parkinson’s and Alzheimer’s diseases [35].

In this study, we tested the impact of changing the naïve level of DJ-1 by suppressing its expression in HEK293 cells. We performed RNA-seq global analysis to assess the role of knockdown using siRNA for DJ-1 levels. We show that siRNA causes a stress response that is converted to an antiviral cascade. The lack of DJ-1 in cells causes a failure in redox homeostasis and the coordinated activation of the STAT pathways that connect a transcriptional wave of an interferon-based antiviral response and a direct effect on mitochondria and RNA metabolism. The potential role of targeting DJ-1 in activating the antiviral response in metastatic cancers is discussed.

## 2. Results

### 2.1. DJ-1 Response to Oxidative Stress by Translocation to Nucleus

PARK7 (DJ-1) was extensively studied for its function as an oxidative stress sensor and in executing the oxidative stress cellular response. This role was associated mostly with neurons and neuronal-like cells that are over-sensitive to oxidation insults (Figure 1A). However, the majority of cell lines express DJ-1 at moderate to high levels (Figure 1B). The current notion is that the modification of DJ-1 on cysteine residue is a prerequisite for DJ-1 nuclear function, including its role in cancer and metastasis.

To this end, we compared the PARK7/DJ-1 gene expression in SH-SY5Y neuroblastoma cells as a model for neuronal-like cells [11], and in HEK293 cells as a non-cancerous cell line. We observed comparable expression levels of DJ-1 transcripts in both cells (Figure 1C). Notably, cell lines exhibit intrinsic capacity to cope with oxidative stress, with neuronal-like cells being more sensitive than other cells’ origins [11,12]. There are 76 antioxidant activity annotated genes (GO: 0016209; Appendix A). Figure 1C shows representative genes, while the expression of DJ-1 is similar in both cell types, many of the genes (e.g., GPX8, GPX4) are highly expressed in HEK293 but display low expression levels in SH-SY5Y. Interestingly, both cell lines expressed the cytoplasmic enzyme SOD1 to different levels, although only SOD2 is expressed in SH-SY5Y cells. SOD2 is located in the mitochondrial matrix and thus impacts mitochondrial apoptosis and the redox function. We anticipate that each cell displays a unique collection of genes to cope with oxidation, rendering a cell-specific capacity for coping with oxidative stress [11].

### 2.2. Knockdown Resulted in 8-Fold Suppression of the Native PARK7/DJ-1 Transcript

We transfected HEK293 cells by siRNA as a negative control (using esiRNA-RLUC, see Section 3) and the siRNA methodology for PARK7 (see Section 3). The cultured cells were collected 24 h following transfection, and total RNA (>200 nt) was prepared for RNA-seq. Figure 2A shows the results of the RT-PCR at two time points (26 h and 48 h, post-transfection). It is evident that already 26 h post-transfection DJ-1 transcripts are strongly suppressed. Quantitation of the degree of suppression by the specific siRNA relative to the non-specific control siRNA showed an 8-fold reduction, which was constant for 48 h. Note that the RULC control showed no reduction in DJ-1 transcripts, supporting the efficiency of the knockdown (KD) protocol used.

Figure 2B shows the results using dimensional reduction by principal component analysis (PCA). Each experimental group is represented by three biological triplicates. The resulting PCA shows the non-treated cells (N.T.), mock transfection with esiRNA-RULC (depicted RULC), and cells transfected with esiRNA-PARK7 (DJ-1). The two major components (PC1 and PC2) explained together 42% of the variance (Figure 2B). The partitioned of the nine samples according to the treatments validates the quality of the results. For the PCA (Figure 2B), we have included all 18k identified transcripts from the RNA-seq results (Appendix A).

Using the results from RNA-seq with biological triplicates, we calculated the extent of the knockdown of PARK7. Only 12% of the original basal levels were detected 24 h after introducing the siRNAs (Figure 2C).

### 2.3. Knockdown of PARK7/DJ-1 Resulted in Coding and Non-Coding Differential Gene Expression

In analyzing all three experimental groups, we observed a large number of differentially expressed genes. The RNA-seq results reported on 18,158 transcripts, 23% of them are non-coding RNAs (total 4192). Figure 3 summarizes DEG findings for the three experimental pairs. Only statistical threshold genes with FDR < 0.05 are included. There are 527 DEGs between KD DJ-1 and the untreated cells (N.T.), and 316 DEGs between cells of KD DJ-1 and cells introduced with non-specific siRNA RULC. Only 92 DEGs were reported between the siRNA of RULC and N.T. cells. Among these reliable results, transcripts that were differentially expressed by log_2_(FC) > 0.5 (i.e., >|1.412| fold) were labeled as upregulated and downregulated transcripts. The numbers of these transcripts are shown in Figure 3 Only a modest number of DEGs were identified between siRNA RULC and untreated (NT) cells (Figure 3); in contrast, a substantial number of DEGs were detected for the siRNA of DJ-1 vs. siRNA RULC (316 genes). We concluded that many DEGs (primarily upregulation set, Figure 3; green) are associated with the effect of the KD of DJ-1, and are not attributed to the unavoidable effects of the siRNA protocol perse. The RNA-seq significant results are listed in Appendix A.

### 2.4. Non-Specific siRNA Induces Interferon Signaling and the Antiviral Response

Upon transfection of the HEK293 with siRNA of RULC, we observed a coordinated induction of genes that are associated with innate immunity and more specifically with interferon response. This observation most likely reflects the non-specific induction of cells to the delivery system for the duplex of RNA molecules and the activation of the dsRNA pathway of viral infection. Testing the induction of RULC vs. naïve, non-treated cells shows the upregulated set of coding transcripts (total of 75 genes, marked interferon stimulation, Figure 4). The network is indicative of the activation of type I Interferons (IFN), which act through the activation of the JAK/STAT signaling cascade to trigger an antiviral response. Note that STAT1 acts as a hub for several transcription factors that belong to the ATF, EGR, and FOS families. The ATF3 is a regulator that links inflammation, oxidative stress, and immune responses [36].

### 2.5. Knockdown of PARK7/DJ-1 Strongly Boosted the Antiviral Response

Figure 5A shows a Venn diagram of the experiment groups. It shows results from comparing the KD of DJ-1 vs. the background of the siRNA of RULC and the non-treated cell. Among the genes that were changed with esiRNA RULC negative control, 45 of them were also induced by using esiRNA for DJ-1.

Figure 5B shows an MA plot of the differential expressed transcripts. Highlighted are set genes that meet the threshold for FDR and FC. Inspection of these genes emphasizes the occurrence of viral-like induction that spans all levels of expression (*y*-axis). Notably, many of the genes were strongly induced (>10-fold). Among the genes with the strongest induction are the interferon (INF) genes and their modulators. For example, the set of OAS genes that recognize dsRNA as a pathogen-associated molecular pattern (PAMP). The induction of OAS1, OAS2, and OASL, which, as main sensors for viral infections, explains the induction of numerous interferon-induced proteins (e.g., IFIT1, IFIT2, IFI44, MX1, MX2). This coordinated signature argues for further amplification of the antiviral response following the KD of DJ-1. Figure 5C shows the strong connectivity of the induced genes using STRING analysis. The analysis used a filtered set by removal of the non-specific set (45 genes, as in Figure 5A) and included the rest of the 275 genes that were upregulated as a result of KD by the siRNA of DJ-1 (Appendix A).

For partitioning the DJ-1 dependent transcriptomic signature from that of the siRNA response perse (i.e., siRNA RULC), we also show in a unified network the genes identified by the siRNA RULC and the KD DJ-1 set (Appendix A). The sub-networks that are unique to cells of the KD of DJ-1 are highlighted. The DJ-1 induced the protein–protein interaction (PPI) network (Figure 5C and Appendix A) is composed of several components: (i) The presenting antigen, which includes genes such as TAP1 and TAP2 that together form the TAP protein complex (transporter associated with antigen processing), along with numerous human leukocyte antigen (HLA) genes of the major histocompatibility complexes (MHC). (ii) The histone with H2 and H4 genes that are indicative of epigenetic, gene expression, and chromatin reorganization. (iii) The main component of interferon and the innate immune response. The hub proteins include the STAT3, STAT2, IRF1, and INFB1 genes. Specifically, IRF1 (interferon regulatory factor 1) is a transcription factor and activator of interferons alpha and beta and gamma. (iv) Components of the immunoproteasome (e.g., PSME1, PSMB9) that are induced by γ-INF. The interferon-γ induces apoptotic mechanisms, nitric oxide production, and antigen-presenting pathways (i.e., MHC class I and II with numerous HLA genes). In addition, IRF1 plays a role in apoptosis and tumor suppression. The result of the activation of type I interferon is shown in the network (Figure 5C) by the increase in inflammatory cytokines and chemokines. Note the activation of smaller connected components such as the BTN (butyrophilin) gene set, which belong to the major histocompatibility complex (MHC)-associated genes, and genes from the classical pathway of the complement system (e.g., C1, C2).

### 2.6. Knockdown of PARK7/DJ-1 Exposed Downregulation of Genes with Membrane-Associated Functions

We identified a small set of genes that was substantially downregulated along with the KD of DJ-1 (57 genes). There is no sign of the involvement of the innate system among the downregulated genes (Figure 6A). We note that the degree of downregulation is modest. Inspecting the network shows that on average, the node degree is low, even at a relaxed threshold of PPI confidence (score > 0.4). The gene that was maximally downregulated is CERK (ceramide kinase; 2.2-fold; Appendix A). CERK is associated with ceramide metabolism through the effect of ceramide and sphingolipids on mitophagy [37]. Another downregulated gene is GET1, which determines the kinetics of mitochondrial targeting proteins, therefore also controlling mitophagy [38]. Surprisingly, among the downregulated genes (Appendix A), many were assigned to neurons (NDNF, neuron-derived neurotrophic factor; NANOS1, nanos C2HC-type zinc finger 1) and synapses, including synaptic vesicle proteins (e.g., synaptophysin, SYP). We attribute this observation to the role of these genes in membrane trafficking. For example, syntaxin 6 (SNX6) functions as an endosomal organizer and intracellular protein transport. Another neuronal-like gene is SYT16, a calcium-independent membranous protein involved in the trafficking of secretory vesicles in non-neuronal tissues. Other genes were implicated in controlling neuronal survival and differentiation (e.g., GFRA1, a member of the glial cell-line-derived neurotrophic factor (GDNF) receptor family), and cellular adhesion (CNTNAP2, which mediates interactions between neurons and glia during development). Other genes belong to the TNF-receptor superfamily (e.g., TNFRS10D, TNFRSF21), indicative of their role in inflammation and in inhibition of cell apoptosis.

Figure 6B indicates the set of genes that can be attributed to the KD of DJ-1 after filtering out the 206 upregulated overlapping genes (inset, Venn diagram). We focused on the 129 upregulated DJ-1 gene set (Figure 6B). These genes were defined as significant (PDR < 0.05) among coding genes that showed a minimal log (FC) > 0.5). We identified STAT3 as a hub that connect stress, inflammation with transcription regulation (e.g., EGR1 and ATF3), and the DNA repair and cell cycle axis (CDKNA1 and DDB2). By eliminating the overwhelming signature of antiviral response, we could highlight the smaller gene set that was induced. As an example is the set of ANKRD36 family members, which was implicated in the pathology of immune and metabolic diseases [39], and considered as a potential diagnosis marker of Chronic myeloid leukemia [40].

## 3. Methods

### 3.1. Cell Lines and siRNAs

Human neuroblastoma SH-SY5Y cells (CRL-2266) were purchased from ATCC (American Type Culture Collection, Rockville, MD, USA). Cells were cultured in Minimum Essential Media (MEM and F12 ratio 1:1, 4.5 g/L glucose) with 10% fetal calf serum (FCS) and 1:10 L-Alanyl-L-Glutamine. Cell cultures were incubated at 37 °C in a humidified atmosphere of 5% CO_2_.

HEK293 cells were cultured to reach a 20–30% confluent level (3 × 10^4^ per cm^2^). For transfection, we used Lipofectamine 2000 (Lipo2000; Invitrogen, Cat # 11668019, Carlsbad, CA, USA) in cells at 40–50% confluency according to the manufacturer’s instructions. Lipo2000 was shown to be ideal for HEK293 [41]. The esiRNA (Mission system, Sigma-Aldrich, Burlington, MA, USA) consists of a pool of hundreds of siRNA (21 bp each), where each individual dsRNA has a low concentration in the pool, which diminishes most off-target effects, while producing an efficient knockdown. We apply the esiRNA-RULC of a mixture of 21 nt dsRNA used as a negative control (RLUC stands esiRNA directed against Renilla Luciferase; EHURLUC) in addition to the specific PARK7 (to knockdown the DJ-1 transcripts and protein products; EHU113961). RULC experiments were used to measure the baseline effects of introducing esiRNA that controls for the delivery method and allows to distinguish the cellular response to the targeted siRNA itself [42].

### 3.2. Reverse Transcription PCR (RT-PCR) and PCR

Cells were collected for RNA preparation. Total RNA was extracted from cell culture with TRIzol (Thermo-Fisher Scientific, Waltham, MA, USA), and RT was performed using a Ready-To-Go first-strand synthesis kit (Cytiva, Marlborough, MA, USA) according to the manufacturer’s instructions. RNA was reverse transcribed into cDNA (1 μg) and used in the PCR reaction. The PCR conditions consisted of denaturation at 95 °C for 2 min and 35 cycles (10 s at 95 °C, 15 s at 60 °C, 30 s at 72 °C), and 5 min for a final extension. The PCR products were separated on 1.5% agarose gel and stained with ethidium bromide, followed by densitometry measurement (using image processing ImageJ program Ver 1.54, from GitHub). The primers were designed against human RefSeq by the Primer3Web tool (Ver 4.1.0). The forward (F) and reverse (R) primers of β-actin were F: CATGCCCACCATCAGCCCTGG and R: ACAGAGCCTCGCCTTTGCCGA, respectively. For DJ-1 (376 nt), the primers were F: GCCTGGTGTGGGGCTTGTAA and R: GCCAACAGAGCAGTAGGACC. Amplicon size was confirmed to be 196 nt and 376 nt for β-actin and PARK7, respectively.

### 3.3. RNA-Seq

Total RNA was extracted using the RNeasy Plus Universal Mini Kit (QIAGEN, Cat # 73404, Redwood City, CA, USA), according to the manufacturer’s protocol. Total RNA samples (1 μg RNA) were enriched for mRNAs by pull-down of poly (A). Libraries were prepared using a KAPA Stranded mRNA-Seq Kit, according to the manufacturer’s protocol, and sequenced using Illumina NextSeq 500 to generate 85 bp single-end reads (a total of 25–30 million reads per sample).

### 3.4. Bioinformatic Analysis

Next-generation sequencing data underwent quality control using FastQC, version 0.11.9. They were then preprocessed using Trimmomatic (Ver. 0.32) and aligned to the reference genome GRCm38 with the STAR aligner (Ver. 2.7.0a) using default parameters. Genomic loci were annotated using GENCODE (release 46). Genes with low expression were filtered out of the dataset by setting a threshold of a minimum of two counts per million in three samples.

Differential analyses were performed on all experimental groups, and genes with an FDR < 0.05 were considered. Only genes with an absolute log fold change of ≥0.5 were labeled up- or downregulated, all the rest were considered unchanged. We partition genes by type as coding and non-coding (including pseudogene, anti-sense, miRNAs, TEC, lncRNA, and other rare biotypes).

RNA-seq experiment results are visualized by an MA plot that transforms the data into M (log ratio) and A (mean average) scales. The functional analysis and network view was based on STRING protein–protein interaction (PPI) platform. The minimal PPI confidence score ranges from 0.4 to 0.9. For protein interaction, we collected data from a human-centric view from IntAct, limiting DJ-1 interacting proteins to a minimal MI score > 0.6. Only evidence of physical interactions was included [43]. We used the gene expression normalization nTPM to compare the expression of DJ-1 across cell types. Data are available in Human Proteome Atlas (HPA) [44].

### 3.5. Statistical Analysis

Principal component analysis (PCA) was performed using the R-base function prcomp (R studio Ver. 4.1.0). EdgeR (Ver. 3.36.0) was used to perform RNA read counts by the trimmed mean of the M-values normalization of RNA (TMM) and differential expression analysis [45]. Figures were generated using the ggplot2 R package (Ver.3.3.5) [46].

## 4. Discussion

In this study, we investigated the role of KD in DJ-1 in the cellular context of HEK293 cells. We identified a robust and strong antiviral-like response that was also moderately induced by using the non-specific siRNA of RULC. The use of siRNA methodologies across different cell lines is a variable that is poorly controlled. Most commercially used siRNA protocols rely on a combination of a small set of specific sequences (about four), while the negative control setting uses scrambled sequences to ensure specificity. Several systematic studies indicated the importance of the cell-type-dependent nature of sensitivity to siRNA by its length and its impact on gene expression related to the innate immune response. For example, in HeLa cells, introducing dsRNAs longer than 23 bp induced toxicity and cell death, while shorter RNA fragments did not affect viability. In all cell types, short siRNA duplex (19 nt) did not affect cell viability [47].

It is known that siRNA delivery reagents may induce stress in the subjected cells. Specifically, Lipo2000 is characterized by high transfection efficiency. As a byproduct of the use of Lipo2000 for siRNAs, autophagy is induced to cope with cellular oxidative and ER stress signals [48]. We showed a strong induction of the antiviral-like response in HEK293 with no sign of cytotoxicity, and we still achieved effective silencing, retaining only 12% of the basal level of expression following 24–48 h post-transfection of siRNAs. The siRNA system and delivery are also sensitive to the presence of sequence motifs via the activation of toll-like receptors (e.g., TLR8) to induce an undesirable antiviral response [49]. Due to the importance of siRNAs for research and as a promising therapeutic method, optimization is needed to achieve the desired responses while optimizing silencing efficiency.

Similar to our finding, siRNAs (21 nt) that were introduced into cells for silencing specific genes, triggered the JAK/STAT signaling pathway [50]. The signaling cascade was initiated by EIF2AK2 (PKR, an IFN-induced dsRNA-dependent kinase), which plays a central role in the innate immune response to viral infection [51]. In our experiments, EIF2AK2 was already overexpressed (four-fold) by the siRNA of the RULC, which is likely to mediate the observed antiviral-like response. The involvement of trafficking and organelle biogenesis in the antiviral response is established, along with the executing of INF signaling [52].

It was shown that DJ-1 deficiency or knockdown in primary cortical neurons and mouse embryonic fibroblasts caused alternations in mitochondrial shape [53]. Knockout mice and DJ-1 pathogenic mutations led to elevated ROS production, lower mitochondrial membrane potential, and changes in gene expression, which can thus explain the importance of DJ-1 function in neurodegenerative diseases [54]. These in vivo findings corroborate the observed effects of suppressing DJ-1 on trafficking, mitophagy, and redox homeostasis (Figure 6A).

We observed that the antiviral effect was strongly amplified by the KD of DJ-1 (with ~200 overexpressed genes and ~100 non-coding RNAs). Most of the ncRNA DEGs are expressed in a moderate-to-low level (Appendix A). While no function is known for most of these non-coding RNAs, the upregulation of lncRNA of NEAT1 (2.2-fold, *q*-value 3.7 × 10^−5^) is intriguing, as NEAT1 was proposed to be upregulated in response to oxidative stress [55] and as a potential biomarker in Parkinson’s diseases [56]. We questioned whether the effect of the KD of DJ-1 is through the signaling convergence of the IFN signaling pathways or, alternatively, through the failure of the cells to cope with the lack of DJ-1 protection and changes in the redox level. We suggest that it is the former explanation. We have not challenged the cells with an oxidative stimulus [11], and, as anticipated, no sign of cytotoxicity following siRNAs was observed, which argues for uninterrupted mitochondrial homeostasis. In contrast, the functional networks presented in this study (Figure 4, Figure 5, Figure 6 and Appendix A) indicate the key role of JAK/STAT as the main hub. DJ-1 negatively regulates inflammatory responses in astrocytes and microglia by facilitating interactions between STAT1 and its phosphatase, SHP-1 [30]. Experiments performed on DJ-1 knockout mice led to increased inflammatory mediators compared to wild-type mice. This was explained by the direct interaction of DJ-1 with SHP-1 and shifting the levels of phosphorylated STAT1. Therefore, DJ-1 may prevent excessive STAT1 activation and reduce the risk of brain inflammation [30]. It remains to be tested whether in our cell system, the activation of the strong antiviral response in KD DJ-1 cells is mediated through such a signaling pathway.

In this study, we tested the KD of DJ-1 in a non-cancerous established cell line by inspecting the transcriptional response. We emphasized that different cells present varying sensitivity to molecular perturbations and external stimuli [57]. We argue that the depletion of DJ-1 in HEK293 cells caused a slight alteration in membranous trafficking (Figure 6B) but mostly a change in transcription, rendering the cells labile and prone to multiple stressors, including dsRNA-INF-dependent viral-like stress. Our study highlights the intricate interplay between siRNA technology and the robust activation of the innate immune system.

## Figures and Tables

**Figure 1 ijms-25-07550-f001:**
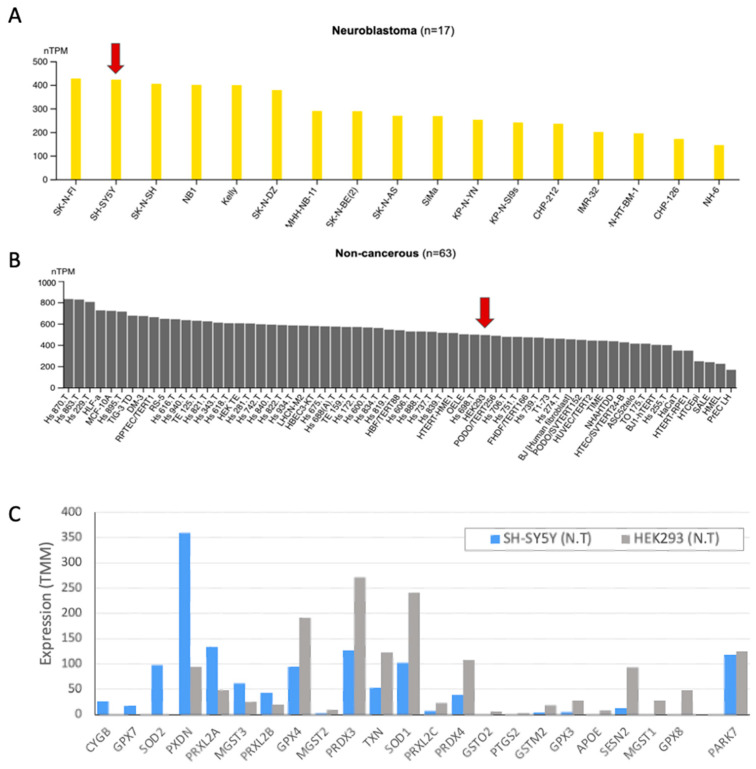
Cell lines characterization in view of DJ-1 and antioxidant activity: (**A**) PARK7 expression levels extracted from neuroblastoma and non-cancerous cell lines. Experimental data were normalized and harmonized to allow confidence in comparing different cells within and across groups (using nTPM). (**A**) The expression levels of the SH-SY5Y among neuroblastoma cell lines (17 cells, 423 nTPM). Red arrow marks the cell of interest. (**B**) PARK7 expression levels in HEK293 is 493.5 nTMP. Substantial difference is recorded in PARK7 expression levels among the 63 non-cancerous cell lines. Red arrow marks the cell of interest. Data were extracted from HPA (see Section 3). (**C**) Gene expression in cell cultures of SH-SY5Y (gray) and HEK293 (blue) cell lines for selected genes associated with antioxidant activity (76 genes, GO: 0016209). Full list is available in Appendix A.

**Figure 2 ijms-25-07550-f002:**
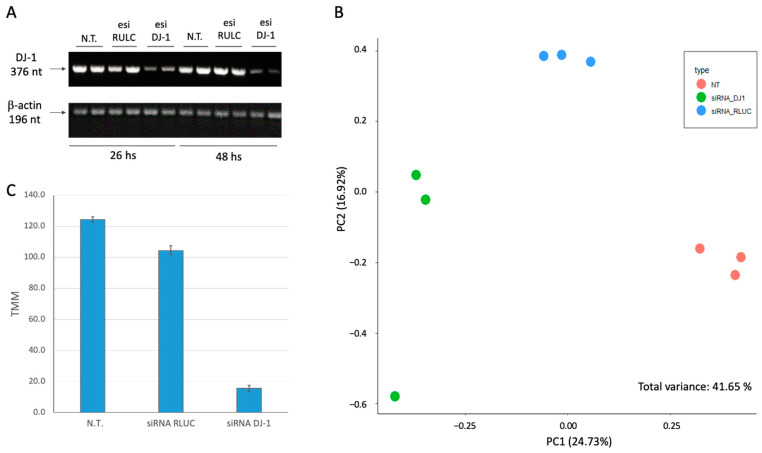
Knockdown of PARK7/DJ-1 using siRNA: (**A**) RT-PCR of the cells prior to treatment, following RULC and PARK-7 siRNA. PCR amplicons were separated by agarose gel separation. Each sample was compared to β-actin, a normalization control (196 nt). The siRNA was tested for 26 and 48 h following transfection. (**B**) PCA for 9 cell samples, based on the top 1000 differentially expressed genes (DEGs) colored by the three experimental groups: the non-treated cells (N.T.), negative control transfection with esiRNA-RULC (RULC), and cells transfected with esiRNA-PARK7 (DJ-1). Each sample is represented by a colored dot. The variance explained is indicated for PC1 and PC2. The explained variance of PC1 and PC2 reached 41.6%. (**C**) The results from B for the level of expression of DJ-I are shown. Each sample was normalized by TMM methods for 1 M reads accounted for 18,158 transcripts.

**Figure 3 ijms-25-07550-f003:**
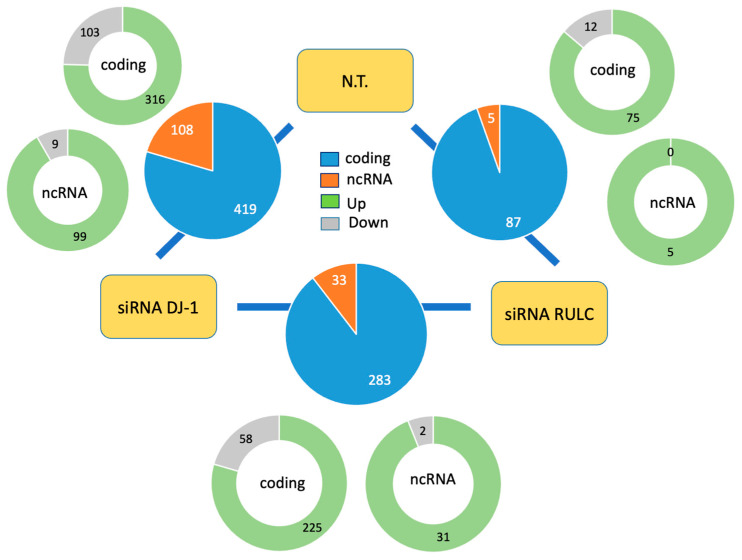
Quantitative summary of the differentially expressed transcripts of DJ-1 siRNA relative to control.. Partition of the significant differentially expressed genes (DEGs) that meet the threshold of FDR < 0.05 and are marked as Up- and Downregulated (log_2_(FC) > |0.5|) for the pairs of cellular settings. Each analysis shows the number of DEGs according to coding (blue) and noncoding RNAs (orange). Partition of the DEGs to those that are upregulated (Up, green) and downregulated (Down, gray) are shown for each of the subsets of protein-coding and non-coding sets. N.T., non-treated cells.

**Figure 4 ijms-25-07550-f004:**
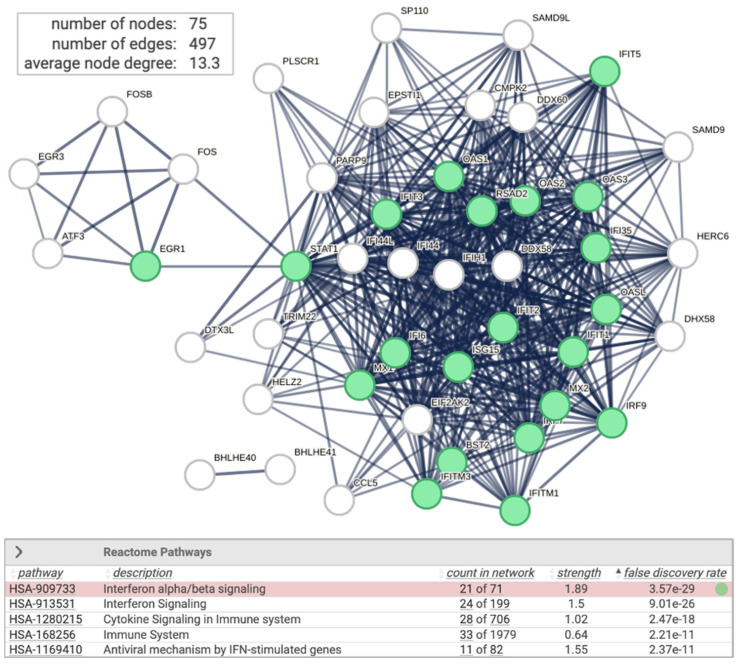
Network of interferon signaling induced by non-specific siRNA: STRING view for 75 significantly upregulated DEGs (FDR < 0.05 and log_2_(FC) > 0.5) from analysis of siRNA RULC vs. non-treated (N.T.) cells. STRING PPI confidence is >0.9 (top). Several significant Reactome pathways are enriched (bottom). The nodes in the graph of Interferon alpha/beta signaling (HAS 909733) are colored green. The other enriched pathways (bottom) are strongly connected to the immune system and antiviral IFN-induced mechanism (uncolored).

**Figure 5 ijms-25-07550-f005:**
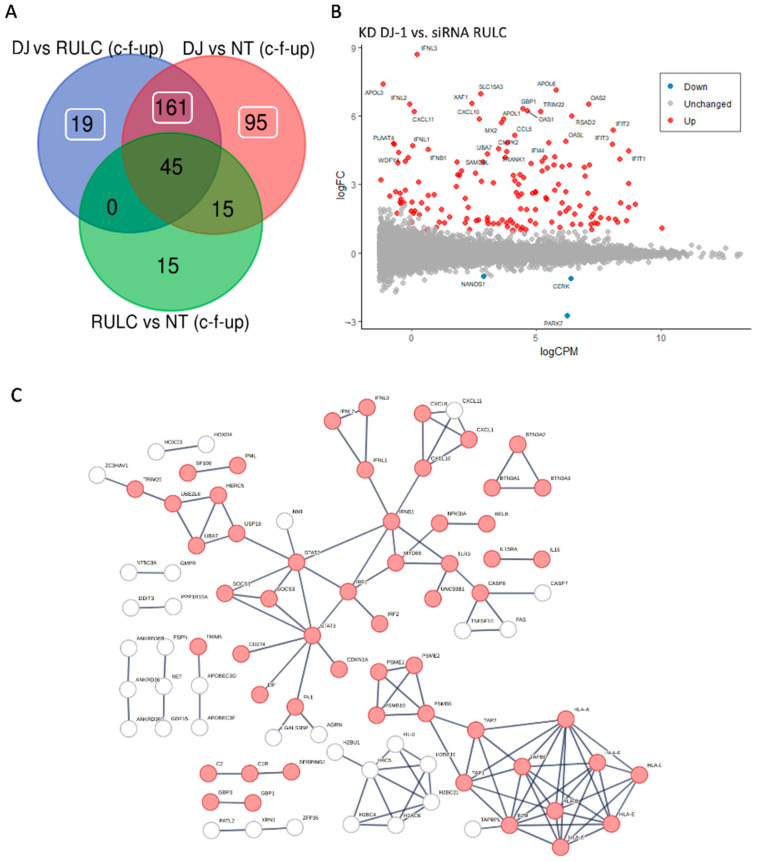
Differentially expressed transcripts of DJ-1 siRNA relative to control: (**A**) A Venn diagram of the experiment groups. The number of overlapping genes is indicated for comparing KD of DJ-1 vs. siRNA of RULC and vs. non-treated cell. The mark of c-f-up indicates the use of filters by coding genes (c), FDR (<0.05, f) and only upregulated DEGs (up). (**B**) MA plot is shown as the log_2_(FC) for each gene (*y*-axis) vs. its mean expression between the two groups as a log_2_(CPM) (counts per million reads; *x*-axis). DEGs included refers to KD of DJ-1 compared to siRNA-RULC. Blue and red points indicate genes that are downregulated and upregulated, respectively. (**C**) STRING view for 275 significantly upregulated DEGs (FDR < 0.05 and log_2_(FC) > 0.5) from analysis in which all 75 genes identified by the siRNA-RULC vs. non-treated (N.T.) cells were omitted. STRING PPI confidence is >0.9 and the *p*-value for PPI enrichment is <1.0 × 10^−16^. The light red color marks immune response genes (GO cellular process: immune system process). The network includes only DEGs that meet the thresholds of FDR < 0.05 and log_2_(FC) of >0.5, and are connected with ≥2 genes.

**Figure 6 ijms-25-07550-f006:**
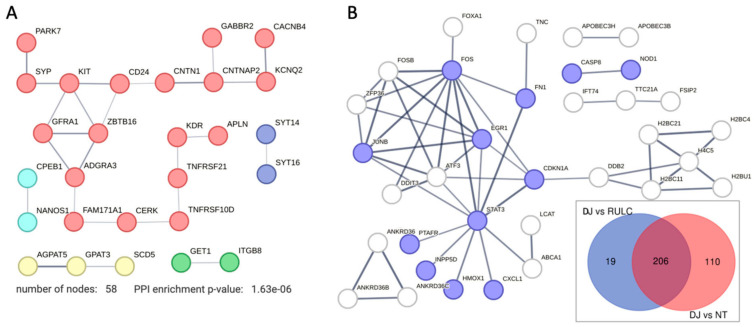
DJ-1 dependent effects on DEGs: (**A**) STRING-based network for all genes that were significantly downregulated with threshold of FDR < 0.05 and log_2_(FC) < −0.5. STRING PPI confidence score > 0.4. (**B**) STRING-based network for 129 genes that were significantly upregulated with threshold of FDR < 0.05 and log_2_(FC) > 0.5. In inset, a Venn diagram of KD of DJ-1 relative to N.T. cells (red), compared to siRNA-RULC relative to N.T. cells (blue). The analysis was performed on the unshared 19 and 110 genes. STRING PPI confidence score > 0.7. Genes enriched by Reactome pathway HSA-1280215: cytokine signaling in immune system (*q*-value 0.0097) are colored purple.

## Data Availability

RNA-seq data files are available through the Appendix A.

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
