# Peer review of "Knockdown of DJ-1 Resulted in a Coordinated Activation of the Innate Immune Antiviral Response in HEK293 Cell Line"

_ijms, 2024, doi:10.3390/ijms25147550_

Round 1
Reviewer 1 Report
Comments and Suggestions for Authors
This manuscript deals with the function of DJ-1 (PARK7) in the cell line HEK293.
First, the authors find that DJ-1 translocates into the nucleus upon treatment with H2O2.
A bioinformatic analysis of HEK293, in which the expression of DJ-1 was switched off, revealed that this molecule plays a role in the antiviral response.
I think this manuscript is questionable in several respects.
1- The experiments on translocation of DJ-1 molecules induced by H2O2 treatment were performed in SH-SY5Y neuroblastoma cells. Based on the images shown in the paper, it cannot be determined that some DJ-1 molecules were actually translocated to the nucleus. The authors should present convincing experiments on this point.
2- It is not clear why the authors did not perform the same experiments in HEK293 cells. This is necessary to confirm that DJ-1 shows similar behaviour in these two cell lines.
3- Panel 1C shows that the two cell lines differ in terms of the presence of antioxidant components. In addition, there are no data on the protein expression of these components in the two cell lines. Therefore, the data shown are limited in their biological significance.
4-The KO is shown for DJ-1 gene expression, not for the molecule. WB analysis should be performed to confirm the finding that DJ-1 is reduced.
5- although it is not clearly stated in the text or in the legend to Figure 3, it appears that the KD of DJ-1 resulted in a strong change in coding and non-coding RNAs compared to the control. I assume the control should be the same as in Figure 2, so the difference is compared to something that can actually do something on the target cells according to the chapter 3.4. Please show where the differences are compared to untreated cells.
6- Chapter 3.4 shows what can actually be expected when we treat a cell to silence something. Silencing is indeed similar to a viral infection. But we should remember that it is not a viral infection. Anyway, it is quite difficult to understand what is specific and what is not when DJ-1 KD is performed to change the expression of multiple genes. Please discuss this in detail.
7- If I understood the chapter 3.4 correctly, the authors want to show the differences in activation of IFN-related genes in KD DJ-1 HEK293 cells compared to control (RULC). And in the following figure, the authors show what was downregulated (surface-associated molecules). Actually, the aim of the manuscript is not clear. In the abstract, the authors focused on the effects of DJ-1 in the context of the KD, and they noted several findings. They found that control for silencing can also trigger a response in target cells. At this point, I think it is really difficult to understand what depends on the specific KD of DJ-1 and what does not.
If the authors want to claim that DJ-1 can trigger an antiviral response, they should prove it. Furthermore, there is no evidence that the DJ-1 molecule is actually downregulated. Without this evidence, the data are difficult to interpret.
All considerations about tumour cells have nothing to do with this manuscript and should be deleted. The fact that PARK7 can play a role in cancer has already been shown in many publications.
Mino point
the last sentence of the manuscript is abbreviated.
Comments on the Quality of English Language
English is good enough.
Author Response
Reply: point by point (Reviewer 1)
We thank the reviewer for carefully reading and commenting on our manuscript. Please note that changes are shown in red font.
Based on the comments we incorporated changes. Primarily: Replaced Fig 1 and Fig 3. Added Suppl. Fig. S2 and S3.
1- The experiments on translocation of DJ-1 molecules induced by H2O2 treatment were performed in SH-SY5Y neuroblastoma cells. Based on the images shown in the paper, it cannot be determined that some DJ-1 molecules were actually translocated to the nucleus. The authors should present convincing experiments on this point.
Reply: While translocation of DJ-1 molecules induced by H2O2 in SH-SY5Y cells is a very sensitive measure for oxidative stress response (shown by us and others), we agree that it may have caused a confusion. We have removed this aspect and focused on transcripts of DJ-1 rather than on its protein.
2- It is not clear why the authors did not perform the same experiments in HEK293 cells. This is necessary to confirm that DJ-1 shows similar behaviour in these two cell lines.
Reply: We failed to perform immunofluorescence measurement in HEK293. These cells are known to be a difficult for immunofluorescence (poor adhesion along with repeated washings).
Revised: We removed the original Fig. 1A and 1B and the associated test. Instead, we included (Revised Fig. 1A and 1B,) DJ-1 expression profile across many cell types including neuroblastoma and non-cancerous cell lines (originally was in a Supplementary Fig. S1). We added missing refences in Introduction for the role of DJ-1 in broad cellular contexts.
3- Panel 1C shows that the two cell lines differ in terms of the presence of antioxidant components. In addition, there are no data on the protein expression of these components in the two cell lines. Therefore, the data shown are limited in their biological significance.
Reply: The entire goal of Fig. 1C was to show that cells provide different transcriptomic levels as their baseline capacity to deal with oxidative stress. As the rest of the paper focuses only on gene expression, it can help the reader to appreciate ‘cell-centric’ view for oxidation response capacity.
4-The KO is shown for DJ-1 gene expression, not for the molecule. WB analysis should be performed to confirm the finding that DJ-1 is reduced.
Reply: As mentioned, the manuscript deals with transcript levels in cells following DJ-1 knockdown. While we do not elaborate on this aspect, The half-life of the DJ-1 was estimated to be 6 h (Taira, 2004 and references within). RNA-seq experiment was performed 26 and 48 h following siRNAs without any external stress.
5- although it is not clearly stated in the text or in the legend to Figure 3, it appears that the KD of DJ-1 resulted in a strong change in coding and non-coding RNAs compared to the control. I assume the control should be the same as in Figure 2, so the difference is compared to something that can actually do something on the target cells according to the chapter 3.4. Please show where the differences are compared to untreated cells.
Reply: As suggested we replace Fig. 3 to include all pairs of analysis and showed for each not only the partition of coding to ncRNAs but also the DEG by Up and downregulation. In addition, we forwarded the reader to Supplementary Table S2 that present all findings from the original expression for 18k transcripts along with a description to the nature of them (coding, non-coding). We also provide 3 sheets for DEG (Supplementary Table S3). The allowing the reader to compare any pairs of the experimental settings shown in Fig. 2B.
6- Chapter 3.4 shows what can actually be expected when we treat a cell to silence something. Silencing is indeed similar to a viral infection. But we should remember that it is not a viral infection. Anyway, it is quite difficult to understand what is specific and what is not when DJ-1 KD is performed to change the expression of multiple genes. Please discuss this in detail.
Reply: This is exactly the message. The use of siRNA (the non-specific RULC) induced a moderate but coordinated INF response without the presence of any virus (or other pathogen) around. To avoid confusion, we marked it as antiviral-like response. When the same protocol was applied with cells lacking DJ-1, the IFN reaction was induced at a different scale (200 unique genes in addition to less than 50 non-specific genes).
Revised: To clarify better and improved the visualization assessment, we added a new Supplementary Fig. S2. This network (highly specific with STRING score of 0.9, relying only on major evidence for protein-protein interaction network) marks each gene that was including in the non-specific siRNA setting (yellow frame). The parts of the networks that were not induced by the non-specific activations are shaded in purple. It is evident that the response to KD of DJ-1 is more robust and extensive. We added this information to the revised version with a new Supplementary Figure S2.
7- If I understood the chapter 3.4 correctly, the authors want to show the differences in activation of IFN-related genes in KD DJ-1 HEK293 cells compared to control (RULC). And in the following figure, the authors show what was downregulated (surface-associated molecules). Actually, the aim of the manuscript is not clear. In the abstract, the authors focused on the effects of DJ-1 in the context of the KD, and they noted several findings. They found that control for silencing can also trigger a response in target cells. At this point, I think it is really difficult to understand what depends on the specific KD of DJ-1 and what does not.
Reply: We are sorry for the apparent confusion. The goal of this study was to expose the pretty dramatic and coordinated antiviral-like response (as correctly stated by the reviewer, without a virus). We further showed that through unbiased view of the entire transcriptome, we can uncover the DJ-1 role in its basal state (no external stimuli were included). We claim that the effect on organelles such as mitochondria and trafficking is exposed (mostly by the down-regulated genes) as well as the boosting of an anti-viral-like cell response. We claim that shutting down DJ-1 can be a useful approach to render cells to become labile.
If the authors want to claim that DJ-1 can trigger an antiviral response, they should prove it. Furthermore, there is no evidence that the DJ-1 molecule is actually downregulated. Without this evidence, the data are difficult to interpret.
Reply: The coordinated network of gene expression is associated with extreme statistical enrichment findings. It makes our observation with over 200 transcripts quite robust. We fully agree that in the future it will be of interest to sort out the signaling pathways with respect to DJ-1 in basal and stimulated settings. We added this as a future direction to the discussion.
All considerations about tumour cells have nothing to do with this manuscript and should be deleted. The fact that PARK7 can play a role in cancer has already been shown in many publications.
Reply: Regarding the cancer relation, we did not elaborate on the role of PARK7 in cancer at all. In view of this study, we propose that shutting down DJ-1 might be useful approach to induce cell fragility and thus improve cancer management approach.
Mino point
the last sentence of the manuscript is abbreviated.
Reply: Corrected.
Reviewer 2 Report
Comments and Suggestions for Authors
Due to the fact that mutation of the PARK7 (DJ-1) gene can lead to sporadic forms of Parkinson's disease, the authors in this paper conducted a study on HEK293 cells with DJ-1 suppression. Consequently, they obtained very interesting and valuable research results that fit the profile of the journal and are worth publishing.
Minor comments, are mainly of an editorial nature and are intended to improve the clarity of the paper.
1. lines 11-16 are superfluous and not in accrodandace to journal guidelines.
2. line 123 - replace with "cm2"
3. Line 124 and the rest of MM - reagent data is incomplete. Please indicate the country of origin (possibly city).
4. Line 160 - please describe how the primers were designed. How were they validated?
5. line 193 - firstly, this abbreviation has already been introduced on page 47, secondly, please decide "DJ-1" or "Dj-1".
6. Figure 1A - it would be clearer to add a scale to the pictures.
7. line 384 - the abbreviation CML does not appear again in the text. Therefore, I do not see the point of introducing it.
8. line 398 - the first place where Lipofectamine 2000 appears is on page 125, and that is where the name should be abbreviated.
Author Response
Reply: point by point (Reviewer 2)
Due to the fact that mutation of the PARK7 (DJ-1) gene can lead to sporadic forms of Parkinson's disease, the authors in this paper conducted a study on HEK293 cells with DJ-1 suppression. Consequently, they obtained very interesting and valuable research results that fit the profile of the journal and are worth publishing.
Reply: We thank the reviewer for appreciating the novelty of our study, and for carefully reading and commenting. All changes are shown in red font. Note that according to the request from Reviewer 1, we incorporated quote a lot of changes. Primarily we have replaced Fig. 1 and Fig. 3, and added Suppl. Fig. S2 and S3.
Minor comments, are mainly of an editorial nature and are intended to improve the clarity of the paper.
- lines 11-16 are superfluous and not in accrodandace to journal guidelines.
Removed.
- line 123 - replace with "cm2"
corrected
- Line 124 and the rest of MM - reagent data is incomplete. Please indicate the country of origin (possibly city).
Completed
- Line 160 - please describe how the primers were designed. How were they validated?
We added the needed information.
- line 193 - firstly, this abbreviation has already been introduced on page 47, secondly, please decide "DJ-1" or "Dj-1".
It is DJ-1 throughout. We kept this naming PARK7 (DJ-1) to refresh this as we consider it exchangeable.
- Figure 1A - it would be clearer to add a scale to the pictures.
This picture was removed (according to request of Reviewer 1).
- line 384 - the abbreviation CML does not appear again in the text. Therefore, I do not see the point of introducing it.
Indeed, removed.
- line 398 - the first place where Lipofectamine 2000 appears is on page 125, and that is where the name should be abbreviated.
Corrected.
Round 2
Reviewer 1 Report
Comments and Suggestions for Authors
The authors deleted part of Figure 1 in which it was questionable the translocation to the nucleus of the PARK7 after H2O2 treatment. Most data were derived from a bioinformatic analysis. There is no clear association with the expression of the proteins considered. The actual protein disappearance of PARK7 factors is not shown, at least from what I can see.
However, in this form, the message of the manuscript is less confusing, although questionable. This paper can be accepted, although tables are less readable than figures.
Comments on the Quality of English LanguageEnglish is good
Author Response
Reviewer 1.
As mentioned - we have removed the protein aspect from the manuscript to make it simpler and more focused. There are quite a lot of cases in which experiments were done by following the level of DJ-1 proteins in cells under basal and oxidative conditions. We did not elaborate on these aspects and made the essential changes in the manuscript to present a simpler and focused report on the anti-viral-like response in HEK293 cells.
We thank the reviewer for the excellent suggestions for improving the manuscript.
We have also edited to improve writing and clear a few typos.